# Protocol for a longitudinal, prospective cohort study investigating the biology of uterine fibroids and endometriosis, and patients' quality of life: the FENOX study

Thomas Theodor Tapmeier [ORCID],[1] Hannah Mohamed Nazri,[1] Kavita S Subramaniam,[1] Sanjiv Manek,[2] Kurtis Garbutt,[1] Emma J Flint,[1] Cecilia Cheuk,[1] Carol Hubbard,[1] Kelly Barrett,[1] Emily Shepherd,[1] Krina T Zondervan,[1,3] Christian Malte Becker[1]

¹Nuffield Department of Women's & Reproductive Health, University of Oxford, Oxford, UK
²Department of Cellular Pathology, Oxford University Hospitals NHS Foundation Trust, Oxford, UK
³Wellcome Centre for Human Genetics, Oxford, UK

**Correspondence to**
Dr Thomas Theodor Tapmeier;
thomas.tapmeier@wrh.ox.ac.uk

## ABSTRACT

**Introduction** Millions of women suffer from the consequences of endometriosis and uterine fibroids, with fibroids the cause for over 50% of hysterectomies in the USA, and direct costs for their treatment estimated at between US$4 and US$9 billion. Endometriosis commonly affects millions of women worldwide predominantly during reproductive age, with severe menstrual and non-menstrual pain and subfertility the main symptoms. Due to the 'unhappy triad' of endometriosis—lack of awareness, lack of clinically relevant biomarkers and the unspecific nature of symptoms—women wait on average for 8–12 years before the definitive endometriosis diagnosis is made. Treatment options for both conditions are not satisfactory at the moment, especially with a view to preserving fertility for the women and families affected. In the Fibroids and Endometriosis Oxford (FENOX) study, we combine the investigation of fibroids and endometriosis, and plan to collect high-quality tissue samples and medical data of participants over a time frame of 5 years after surgical intervention.

**Methods and analysis** Biological samples such as blood, saliva, urine, fat, peritoneal fluid and—if found—endometrial tissue or fibroids as well as detailed clinical and intraoperative data will be collected from women undergoing surgery and participating in the study after informed consent. We plan to recruit up to 1200 participants per disease arm (ie, endometriosis and uterine fibroids) over 5 years. Participants will fill in detailed and validated questionnaires on their medical history and quality of life, with follow-ups for 5 years. Enrolment started on 2 April 2018, and FENOX will close on 31 March 2028. We will analyse the biological samples using state-of-the-art molecular biology methods and correlate the findings with the medical records and questionnaire data.

**Ethics and dissemination** The findings will be published in high-ranking journals in the field and presented at national and international conferences.

**Trial registration number** ISRCTN13560263.

### Strengths and limitations of this study

► Fibroids and Endometriosis Oxford (FENOX) combines the study of endometriosis and uterine fibroids to identify the underlying mechanisms of both conditions.
► The study comprises biological samples as well as comprehensive phenotypic data.
► World Endometriosis Research Foundation Endometriosis Phenome and Biobanking Harmonisation Project criteria are applied to ensure the best possible standardisation.
► The longitudinal aspect is an important feature of FENOX, but this depends on uptake and compliance by participants.
► The control group comprises women undergoing surgery for gynaecological indications other than endometriosis or uterine fibroids; thus, they are not completely healthy controls.

## INTRODUCTION

Millions of women suffer from the consequences of endometriosis[1 2] and uterine fibroids.[3 4] These include pelvic and abdominal pain, abnormal uterine bleeding, infertility and miscarriages.[5–8] As such, these conditions affect women and their families in their everyday lives, and have been shown to have an enormous socioeconomic impact on society, in general. In the USA, fibroids are cited to be the cause for over 50% of hysterectomies,[9] and direct costs for their treatment is estimated between US$4 and US$9 billion.[10]

Clinically relevant, non-invasive diagnostic tests including biomarkers or imaging techniques do not exist for many forms of endometriosis,[11–13] resulting in an average delay in diagnosis of 8–12 years. Current treatment options are associated with significant side

**BMJ**

effects and risks and include hormonal suppression/modification, surgical removal or, in the case of fibroids, embolisation and MRI-guided focused ultrasound.

Therefore, there exists a significant unmet clinical need to better understand the underlying mechanisms of these conditions, which will enable us to develop more specific diagnostic tests and will eventually lead to individualised treatment, with fewer side effects and better efficacy. To achieve this goal, it is essential to collect prospective high-quality, standardised clinical and intraoperative data and corresponding biological samples. Our group has been at the forefront of the development of standard operating procedures and questionnaires for endometriosis as part of the World Endometriosis Research Foundation's (WERF) Endometriosis Phenome and Biobanking Harmonisation Project (EPHect),[14–17] and we are planning to establish similar standards in uterine fibroid research.

In the Fibroids and Endometriosis Oxford (FENOX) study, we aim to improve our understanding of the underlying mechanisms of endometriosis and uterine fibroids and their associated symptoms by means of longitudinal observation and laboratory analyses. To achieve this, samples and clinical data will be collected from women undergoing surgery. These samples will be used in state-of-the-art biomedical assays (see section 'Assays') to improve our understanding of the underlying biology of these symptoms in women with endometriosis and/or fibroids, which will lead to a better understanding of the conditions, stratification of patient groups and tailored therapies and the development of novel drug targets and biomarkers for diagnosis and treatment.

## Objectives
### Primary objective
▶ To identify the underlying mechanisms of endometriosis and uterine fibroids and their associated symptoms to improve the outcome of affected women.

### Secondary objectives
▶ To identify novel biomarkers of endometriosis.
▶ To identify clinical subgroups of endometriosis and uterine fibroids.
▶ To understand the genetics underlying these conditions and explore the relevant downstream molecular pathways.
▶ To investigate the relation between the presence of fibroids and the symptoms, for example, abnormal uterine bleeding.
▶ To identify novel drug targets.
▶ To develop models of disease progression and prediction.
▶ To investigate conditions or symptoms associated with endometriosis and/or uterine fibroids, including: symptoms and characteristics of the female reproductive system (characteristics of menstrual bleeding, fertility, infertility, pregnancy outcomes), pelvic as well as non-pelvic pain conditions, metabolic phenotypes

(polycystic ovarian syndrome, obesity and fat distribution), cardiovascular conditions and symptoms, neuroangiogenesis and related neurological symptoms, immunological disorders and cancers.

## Outcomes
### Primary outcome
We will used questionnaire data, medical records and sample analysis to investigate the genetic and molecular basis of the pathogenesis and symptoms of endometriosis and uterine fibroids. At the end of the recruiting period, that is, from December 2022 onwards, the collected data and samples will be analysed and compared between endometriosis/fibroid cases and non-affected controls.

### Secondary outcomes
Prospective standardised questionnaires and samples will be collected according to EPHect standards. The correlation of cellular, molecular and genetic data and endometriosis status will allow us to define novel biomarkers of the disease.

Clinical notes and questionnaires in combination with sample data will be used to define clinical subgroups of patients.

The molecular and genetic findings will be compared against public databases of disease-relevant molecular pathways, and in vitro experiments will be carried out to test hypothetical connections between the genetics and manifestation of disease.

The blood vessels and endothelial cells will be compared between tissue from women presenting with fibroids and those without.

The detailed comparison between tissue from women with fibroids and those without will yield differences in terms of proteins expressed; these can then be tested as targets using known or new drugs.

As data accumulate and genetic mechanisms become clear, hypotheses will be formed as to the likely progression of disease. These will be tested against the reports from the follow-up questionnaires.

We will use questionnaire data, medical records and sample analysis to investigate the genetic and molecular basis of the pathogenesis and symptoms of conditions or symptoms associated with endometriosis and/or uterine fibroids.

## METHODS AND ANALYSIS
### Study design
FENOX is a prospective study that aims to improve our understanding of the underlying mechanisms of endometriosis and uterine fibroids and their associated symptoms by means of longitudinal observation and laboratory analyses. Biological samples such as blood, saliva, urine, fat, peritoneal fluid and—if found—endometriosis tissue or fibroids as well as detailed clinical and intraoperative data will be collected from women of reproductive age with and without endometriosis-associated and fibroid-associated

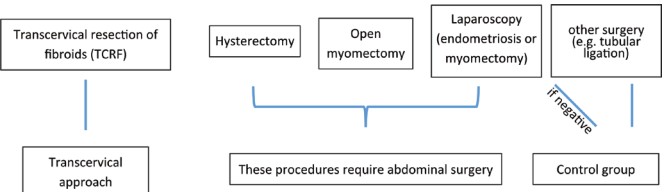

**Figure 1** Participant groups and procedures. The participants will be allocated into case or control groups and tissues will be sourced according to the course of clinical intervention: Participants with fibroids treated by transcervical resection (TCRF) will not undergo abdominal surgery, while abdominal surgery is necessary if the fibroids are treated by laparoscopy, open myomectomy or hysterectomy. In these cases, peritoneal fluid can be obtained in addition to the tissue samples (fibroid, myometrium, endometrium). Endometriosis patients will undergo laparoscopy. If found without endometriosis, they will be grouped with the controls, who are women undergoing surgery for other, unrelated conditions (eg, tubular ligation).

symptoms, such as pain, abnormal uterine bleeding and infertility. Women undergoing surgery for these conditions, and women undergoing surgery for unrelated gynaecological conditions as part of their normal clinical management will be asked to participate (figure 1). An incidental diagnosis of endometriosis or uterine fibroids

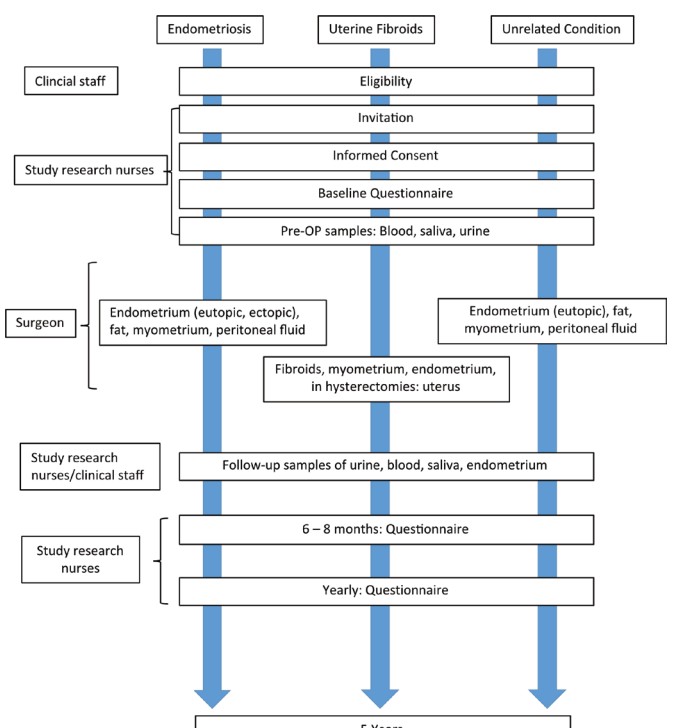

**Figure 2** Flow chart of participant data and tissue sampling. Consented participants of the different arms of the study donate pre-operative samples and fill in a baseline questionnaire. They can donate a range of tissue samples according to their condition and treatment mode (eg, fibroids or endometriotic tissue, peritoneal fluid), and are asked to repeatedly fill in follow-up questionnaires on their condition and quality of life, so that the clinical findings can be correlated with the outcome years later.

will lead to the patients' inclusion into the relevant case groups. All women attending clinics receive a letter informing them of ongoing research, and eligible women will be identified initially by research nurses or clinical staff during clinic visits. Once a woman has expressed an interest in participating in this study, they will be consented by a member of the research team (figure 2).

Blood, saliva and urine will be taken prior to surgery. Tissue and peritoneal fluid (where applicable) will be taken at the time of the scheduled surgery.

In order to determine the effect of the surgical removal of the fibroids on the local tissue, it is necessary to take an additional endometrial biopsy after the planned surgical intervention. This sample will be timed to synch with the same time point in the menstrual cycle that the original sample was taken, and thus will give us a unique insight into the biology of the conditions. During this visit, blood and urine samples will be taken again also. Women can opt in or out of the additional clinic visit where these samples would be taken.

Women will be asked to complete questionnaires on paper, online or into their electronic handheld devices (health, pain, medication and, initially, ethnicity) at different time points. There will be a lengthy questionnaire at baseline before surgery (taking an estimated 45 min to complete), and shorter versions (taking up to 30 min to complete) postoperatively at 6–8 weeks, 6 months, 12 months and thereafter yearly for a total of 5 years after surgical intervention.

### Samples
1. Blood samples (up to 50 mL, venepuncture), urine (micturition) and saliva (spit) will be taken prior to surgery.
2. During surgery, tissue samples will be taken as specified below.
3. In women opting in, an additional endometrial biopsy will be taken during a follow-up visit at least 3 months after surgery. This can be done in an outpatient setting, and the taking of an endometrial biopsy in this setting using an endometrial sampling device (eg, pipelle or curette) is an established technique. A blood and urine sample will be taken again also.

### Participants
In each of the disease arms (endometriosis or fibroids), we plan to recruit up to 1200 women of reproductive age (18 years until menopause) who are planned to undergo surgery. Eight hundred of these will have the condition of interest, and 400 will be having surgery for other reasons and act as controls. In addition, we will include fibroid and uterine tissue samples collected as excess tissue (Oxford Radcliffe Biobank, REC Ref 09/H0606/5+5), currently approximately 70 samples.

### Inclusion criteria
► The participant is willing and able to give informed consent for participation in the study.

► The participant is female and aged 18 years or above (before menopause).
► Women undergoing planned surgery (including hysterectomy) for endometriosis-associated and/or fibroid-associated symptoms such as abdominal pain, abnormal uterine bleeding or for unrelated gynaecological conditions (eg, fertility investigation or for laparoscopic tubal sterilisation).

### Exclusion criteria

The participant may not enter the study if ANY of the following apply:
► Women who are pregnant.
► Women who are unable to read, or to understand written or spoken English.
► History of cancer/diagnosis of current cancer.

### Participant enrolment

#### Recruitment

After general information through a generic letter with information about ongoing research, which every patient will receive prior to her outpatient appointment, eligible women will be identified initially by the research nurses or clinical team during clinic visits. The study research nurses will then contact those women interested in participating in the study.

#### Screening and eligibility assessment

Women attending clinic appointments for endometriosis-associated and fibroid-associated symptoms such as pain, abnormal uterine bleeding and infertility will be asked to participate by clinical staff or by the authorised study research nurses. Women undergoing surgery for these conditions, and women undergoing surgery as part of their normal clinical management (eg, laparoscopic tubal ligation or hysterectomy; they would be the control patients) are eligible to participate in the study.

#### Informed consent

Prior to giving consent, and usually during their preoperative assessment visit, women will be given the relevant patient information sheet and consent form to read. Written consent will be received by a trained member of the research team.

Written versions, with verbal explanations, of the patient information sheet and the consent forms will be presented to the participants detailing the exact nature of the study; what it will involve for the participant; the implications and constraints of the protocol; the known side effects and any risks involved in taking part. It will be clearly stated that the participant is free to withdraw from the study at any time for any reason without prejudice to future care, without affecting their legal rights, and with no obligation to give the reason for withdrawal.

The participant will be allowed as much time as wished to consider the information, and the opportunity to question the Investigator, their general practitioner (GP) or other independent parties to decide whether they will participate in the study.

Written informed consent will then be obtained by means of the participant's dated signature and the dated signature of the person who presented and obtained the informed consent. The person who obtained the consent will be suitably qualified and experienced, and will have been authorised to do so by the Chief Investigator (CI). A copy of the signed informed consent and the patient information sheet will be given to the participant. The original signed form will be retained at the study site. The consent form for this study allows for the participant declining consent for any procedure that she is not comfortable with, while remaining eligible as a participant of the study, for example, if a participant did not want a uterine biopsy used in the study, she would not initial the corresponding box on the consent form, and insert 'no' instead.

### Study settings

#### Baseline assessments

Consented participants will be asked by the research team to complete a baseline questionnaire before their surgery. This may be sent to them for example, via mail or email before surgery or given to them in paper form in the clinic. Alternatively, participants have the option to complete an online version of the questionnaire or they may fill in the questionnaire on their handheld device. Each participant will receive a unique login for the online questionnaires.

The questionnaire data will be withheld from the research team until written informed consent is obtained and will be destroyed if this is not granted.

All participants will be sent further questionnaires at different time points (approximately 6–8 weeks, 6 months, 12 months and then yearly for 5 years after surgery). Participants may be reminded (twice, maximally) to return completed questionnaires via mail, email, phone or text or similar.

On the day of surgery, they will be asked to provide a mid-stream urine and a saliva sample. In addition, blood will be collected by peripheral venepuncture. The procedures will be explained to the women again and they will be given the opportunity to ask questions. Assessment of the presence and extent of disease will be performed by the operating surgeon.

Samples that may be taken at time of surgical procedures are as follows:
1. *Laparoscopy for suspected endometriosis*: endometrial biopsy, peritoneal fluid aspiration, fat tissue biopsy, endometriotic tissue if present, peritoneal biopsy.
2. *Laparoscopy for uterine fibroids*: endometrial biopsy, peritoneal fluid aspiration, fat tissue biopsy, endometriotic tissue (if present), peritoneal biopsy, fibroid tissue, myometrial biopsy.
3. *Laparotomy for uterine fibroids*: endometrial biopsy, peritoneal fluid aspiration, fat tissue biopsy, endometriotic tissue if present, peritoneal biopsy, fibroid tissue, myometrial biopsy.

4. *Transcervical resection of uterine fibroids*: endometrial biopsy, fibroid tissue, myometrial biopsy.
5. *Laparoscopy for tubal sterilisation (controls)*: endometrial biopsy, peritoneal fluid aspiration, fat tissue biopsy, peritoneal biopsy, myometrial biopsy.
6. *Hysterectomy*: whole uterus as excess tissue from women with fibroids as well as from controls without fibroids, who undergo surgery for other indications (such as heavy menstrual bleeding, or pain), peritoneal fluid aspiration, fat tissue biopsy and peritoneal biopsy.

Women undergoing hysterectomy for benign causes such as abnormal uterine bleeding or pain will be asked to donate part of their uterus for research. Hysterectomy specimens are excess tissue and would be discarded otherwise.

During surgery, the surgeon will record digital photographs of the inside of the abdomen and/or uterine cavity as part of routine clinical care, which will also be stored on a secure server identified by the participant's study ID. Intraoperative findings will be recorded by the surgeon and anonymised data collected.

### Subsequent visits
Unless participants underwent a hysterectomy, all included women will be asked to contact the study team when the next menstrual period after the procedure started. Together with the last menstrual period date given at the time of the procedure, this date will be used to calculate the length of the cycle. This is important in order to account for the changes that occur in the uterus during the menstrual cycle, and to enable us to distinguish between the effects of the cycle and the disease.

One subsequent visit will be made by participants treated for fibroids who consent to this. They will have another endometrial sample taken at least 3 months after the surgical intervention in an outpatient setting. The taking of an endometrial biopsy in this setting using an endometrial sampling device (eg, pipelle or curette) is an established technique, takes approximately 30 min and there is only a minimal risk of bleeding. In addition, blood and urine samples will be taken also. Pregnant women will not be eligible for the subsequent visits.

All women will be contacted by a member of the medical or study team and asked to fill in further questionnaires at different follow-up time points (approximately 6–8 weeks, 6 months, 12 months and yearly thereafter for a total of 5 years). Reminders will be sent twice, maximally. Women will also be asked if they can be contacted in the future for any further studies approved by an ethics committee.

### Sample handling
Samples will be obtained according to WERF/EPHect guidelines[14–17] by the interventions listed below. Only the study team will have access. Biological samples will be stored at −20°C or at −80°C for use in current and future studies until exhausted. Disease and control samples will be stored under the same conditions. For the purposes of this study, samples may be analysed at Oxford, or they may be transferred to a third party/study collaborator, including industrial partners, for analysis at their facility. If participants agree, samples will be moved to a Research Tissue Bank at the end of the study or stored and used in future ethically approved studies. They would be made available in anonymised form.

To investigate the relationship between uterine fibroids and symptoms such as abnormal uterine bleeding and pain, we aim to collect endometrial and myometrial samples alongside the fibroids themselves, to be able to detect the effects of fibroids on their surroundings.

We also intend to use the endometrium and endometriotic samples, one of the blood samples and fat samples to look for genetic factors and molecular pathways that can lead to endometriosis or uterine fibroids. The samples for this analysis will also be anonymised so that we do not know specifically which patient they came from. However, all samples are identifiable with printed label and location detail, participant ID, sample type and colour-coded cap.

### Blood
Fifty millilitres. These are divided into (at least) EDTA-treated (2×9 mL) and heparin-treated samples (2×6 mL, both anticoagulation), serum (2×serum-separating tubes (SST), 5 mL) and two plain blood samples of 5 mL. The different vials are colour-coded and frozen at −80°C.

### Urine
Twenty millilitres. Half of the sample will be used to test for glucose by specific gravity assay, the other half will be stored for the study. One aliquot of 5 mL is frozen directly at −80°C; 1 mL is centrifuged at 300 g and 5 aliquots of 200 μL of cell-free supernatant are frozen at −80°C.

### Saliva
A spit sample of approximately 1 mL is taken on ice. One aliquot of 200 μL is frozen at −80°C, the rest is centrifuged at 300 g and 2×200 μL of cell-free supernatant are frozen at −80°C. One aliquot of 50 μL of cell-free supernatant is combined with 200 μL of RNA-preserving buffer (RNA later, Qiagen, Germany) and then frozen at −80°C.

### Peritoneal fluid
During surgery, the peritoneal fluid will be collected by the surgeon using a syringe or through mechanical suction on ice. Depending on the volume (up to 15 mL), an aliquot will be centrifuged at 300 g, and the pellet (cells) stored at −80°C for further analysis. The cell-free supernatant will also be stored at −80°C.

### Endometrium (eg, pipelle or curette), endometrial lesions (peritoneum), abdominal fat, myometrium, fibroid tissue
All tissue will be collected on ice and divided for storage at −80°C and—after fixing in paraformaldehyde and ethanol—at room temperature. Parts of fresh tissues will be used for culturing experiments, in order to test compounds, drugs or similar agents on primary cells.

## Hysterectomy

In agreement with the local pathologist, whole uteri will be taken on ice and used for perfusion experiments within 24 hours before being transferred to pathology. Tissue samples of myometrium, endometrium, fibroid and fibroid-associated vasculature (if present) will be taken and stored at −80°C and—after fixing in paraformaldehyde and ethanol—at room temperature as the other tissue samples above.

## Assays
### RNA analysis

RNA from each sample will be isolated by standard methods. Gene expression studies will be carried out between cases and controls (eg, endometriosis vs non-endometriosis patients, or fibroid-bearing women vs women without fibroids) using quantitative real-time PCR assays, whole RNA sequencing methods and RNA microarrays.

### Protein analysis

Proteins will be extracted from tissue samples using standard methods. The expression and amount of proteins will be analysed by immunoblotting for specific proteins of interest, and by proteomics methods using the matrix-assisted laser desorption/ionisation/surface-enhanced laser desorption/ionisation platform. Tissue sections will be used in standard immunohistochemistry to detect the expression of markers of interest in situ.

### Cells

Fresh tissue will be dissociated into single cell suspensions. From these, the diverse cell types (eg, endothelial cells) will be grown in incubators in vitro in order to study differences in cell behaviour between cases and controls, and to test compounds and drugs. Cells will be analysed by microscopy, flow cytometry and immunocytochemistry methods.

Similarly, cells isolated from peritoneal fluid or blood will be analysed using these methods.

### Secretome analysis (perfusion)

Whole uteri with and without fibroids will be perfused with a suitable buffer for up to 8 hours. The perfusate will be analysed by proteomics methods (see above) to detect factors secreted by the fibroids.

### Microscopy

Tissue blocks (up to 5 mm³ in size) from perfused uteri will be stained with antibodies against markers of blood vessels and fibroids, and leakiness, and be recorded in a confocal microscope in order to render a three-dimensional image of the blood vessels in situ. The detailed study of these will allow us to determine whether there is a significant difference in the architecture of blood vessels in uteri with fibroids compared with those from uteri without fibroids.

### Discontinuation/Withdrawal of participants from study

Each participant has the right to withdraw from the study at any time. In addition, the Investigator may discontinue a participant from the study at any time if the Investigator considers it necessary for any reason including:
► Pregnancy;
► Ineligibility (either arising during the study or retrospectively, having been missed at screening);
► Withdrawal of consent;
► Loss to follow-up;
► Loss of mental capacity.

*Withdrawal from the study*: at the point the participant withdraws from the study, we will ask for consent to retain samples and data collected up to that point. Withdrawn participants will not be replaced. The reason for withdrawal will be recorded in the case record file (CRF).

### Definition of end of study

The end of study is 6 months after the locking of the study database, to allow for completion of data analysis.

### Patient and public involvement

FENOX was built on experience and feedback we received from patients and research nurses during a previous study (a study to identify possible biomarkers in women with Endometriosis at Oxford[18]). In addition, the research objectives were set in accordance with research priorities identified through the James Lind Alliance Priority Setting Partnership (PSP) for endometriosis, in which we participate.[19] The James Lind Alliance brings patients, carers and clinicians together in PSPs to identify and prioritise the top 10 unanswered questions or evidence uncertainties that they agree are the most important.

## INTERVENTIONS
### Non-clinical interventions
#### Questionnaires

Participants will complete specific questionnaires about their condition, general health, pain sensitivity, medication and menstrual history before their surgery. Additionally, those women with endometriosis will be asked to complete the Endometriosis Health Profile (EHP-30) Questionnaire,[20] while women with fibroids complete a section on their quality of life.[21] Currently, the clinical questionnaires are completed by participants on paper; in the future, the aim is that the questionnaires can be completed via a handheld device or via the internet directly onto a secure server. For this, women will receive a pretrial study number and login information. The follow-up questionnaires ask about symptoms and changes in menstrual history as relevant. The control groups would be given the same questionnaires as the women with the respective condition and asked to omit questions not applicable to them.

#### Medical records

We will obtain clinical data (menstrual cycle phase, medication, pain and menstrual bleeding status, photos from surgery) from the patients' medical records.

## Clinical interventions

### Venepuncture

Taken by an appropriately trained member of the clinical or research staff.

### Collection of other bodily fluid sample

Urine and saliva samples donated by the patient and sample prepared and analysed by a member of the investigative team.

### Tissue collection

Tissue/fluid (eg, fibroids if present) will be collected as part of routine surgical management apart from:

### Laparoscopy

Peritoneal fluid will be aspirated, biopsies from endometrium (eg, pipelle or curette), abdominal fat tissue, myometrium and peritoneum (excision) will be taken during surgery.

### Additional risk

Minor bleeding, uterine perforation (<1%), additional length of surgery by 5 min.

### Myomectomy/Hysterectomy

Endometrial, myometrial and/or fibroid tissue biopsies will be taken during surgery. Hysterectomy samples will be used in structural analysis assays ex vivo in close discussion with the clinical pathologists.

*Additional risk*: minor bleeding, uterine perforation (<1%), additional length of surgery by 5 min.

### Transcervical resection of fibroids

Endometrial and myometrial biopsies will be taken during surgery.

*Additional risk*: minor bleeding, uterine perforation (<1%), additional length of surgery by 5 min.

### Additionally

In women opting in, an additional endometrial biopsy will be taken during a follow-up visit in an outpatient setting. The biopsy of the endometrium is a simple, routine procedure and takes about 30 min.

*Additional risk*: minor bleeding, uterine perforation (<1%), short period of discomfort.

Women will be asked to consent to the use of samples and clinical data collected as part of this research and in future research. Women, if they consent, will potentially be contacted for future studies approved by an ethics committee.

## Adverse events

For this study, it is conceivable that additional procedures may result in bleeding. However, if this resulted in a scenario mentioned below, it would constitute a serious adverse event (SAE) and needed to be reported to the sponsor.

An SAE is any untoward medical occurrence that:
► results in death;
► is life-threatening;
► requires inpatient hospitalisation or prolongation of existing hospitalisation;
► results in persistent or significant disability/incapacity;
► consists of a congenital anomaly or birth defect.

Other 'important medical events' may also be considered serious if they jeopardise the participant or require an intervention to prevent one of the above consequences.

*Note*: The term 'life-threatening' in the definition of 'serious' refers to an event in which the participant was at risk of death at the time of the event; it does not refer to an event which hypothetically might have caused death if it were more severe.

An SAE occurring to a participant should be reported to the REC that gave a favourable opinion of the study where in the opinion of the CI the event was 'related' (resulted from administration of any of the research procedures) and 'unexpected' in relation to those procedures. Reports of related and unexpected SAEs should be submitted within 15 working days of the CI becoming aware of the event, using the Health Research Authority (HRA) report of SAE form (see HRA website, https://www.hra.nhs.uk).

## Data analysis plan

As this is a prospective sample and data collection study, there is no randomisation of patients as all women will undergo surgery as part of their routine clinical management. As previously,[18] we will use SPSS, GraphPad Prism, STATA and R for analysis, and employ t-tests, analysis of variance (ANOVA), correlation coefficient analysis and similar methods. We plan to use multivariate logistic regression models in comparisons of endometriosis cases with controls, and fibroid cases with controls, adjusting for confounders relevant to the hypothesis being tested. A priori confounders are likely to be age, ethnicity and menstrual cycle phase. Patients with both endometriosis and fibroids will enter into the analysis according to the research question asked; we will conduct sensitivity analyses on this comorbid group to examine to what extent they influence the results. Due to the exploratory nature of this study, various additional statistical techniques may also be used to fully explore the relationships in the data, but all methods will be fully documented.

## Power calculations

Power calculations were done in R (V.3.6.1) using the *pwr* package. For our primary outcome, we plan to correlate questionnaire and laboratory data. To detect correlations with a moderate effect size of r=0.3 at a power of 80% and 0.05 significance level, we will need 85 samples per group (*pwr.r.test*). For the detection of effect sizes of at least 0.2 between groups (eg, endometriosis cases vs controls, with three cycle phases and five disease stages (0, stages 1–4)) using ANOVA at 0.05 significance and 80% power, at least 32 samples per groups will be used, with a total of 480 samples for all 15 groups (*pwr.anova.test*). Multiple comparisons will be corrected for by Bonferroni's method.

## The number of participants

It is now recognised that both endometriosis and uterine fibroids are very heterogeneous conditions. Our previous studies[22 23] and systematic reviews[24 25] have clearly identified a lack of sufficiently powered studies. Multiple large-scale research collaborations are currently in place investigating different aspects of endometriosis,[26] and we plan similar efforts for uterine fibroids. Therefore, large patient numbers are needed.

The Endometriosis CaRe Centre at Oxford is the UK's largest endometriosis centre. Similarly, as a tertiary referral centre, we see many women with fibroid-associated symptoms. As a result, we have the unique opportunity to collect large amounts of data and samples, which is essential to produce clinically meaningful outputs. Given our current patient recruitment rate (endometriosis: 100/year, uterine fibroids, 200/year), we estimate an enrolment of approximately 2×1200 women over the course of the study (800 endometriosis patients+400 non-endometriotic controls, 800 fibroid patients+400 non-fibroid controls). Fibroids already collected as excess tissue under the Oxford Radcliffe Biobank (REC Ref 09/H0606/5+5) will also be included in this study, currently approximately 70 samples.

## Analysis of outcome measures

All samples excluding those from patients who withdraw consent will be included in the analysis of outcome measures.

Laboratory data will be analysed using assay-specific software packages employing univariate and multivariate pattern recognition methods (eg, principal component analysis, partial least squares, stochastic neighbour embedding algorithms) between sample groups. Correlation with questionnaire data will allow us to validate prospective markers of disease. In addition, we will use laboratory data to predict disease severity (revised American Fertility Society score[27]), quality of life (EHP-30[20]), pain measures and improvement of symptoms as per follow-up questionnaires. For the multivariate predictive methods, a test set of approximately 30% of each treatment group will be selected at random. This may be selected in a stratified method and exclude patients that have particularly extreme values (eg, >3 SD from the mean). Patients not included in the test set will make up the training set. Models will then be built on the training set and assessed for predictability on the test set.

The final analysis will be performed on the whole data set. However, if some influential differences, for example, in body mass index or comorbidities are seen, then for example, the women with endometriosis will be matched to corresponding women without endometriosis, or women with fibroids to women without fibroids, and the analysis based on these matched pairs.

## DATA MANAGEMENT
### Access to data

Direct access will be granted to authorised representatives from the Sponsor and host institution for monitoring and/or audit of the study to ensure compliance with regulations.

## Data recording and record keeping

Each participant will receive a unique study number, which will then be used throughout the study. A study master sheet linking patient identifiable data (name, date of birth, hospital and National Health Service numbers) with the unique study number will be kept and password protected on the University of Oxford's High Compliance server with authorised access and in a file separate from the main study file. Hard copy study documents will be kept in a locked room at each participating centre. Research data will therefore be using non-identifiable data, and all records will be identified only by this study number. All study data will be entered on a desktop computer into a program such as Microsoft EXCEL or Sapphire (Labvantage) using password protection. The participants will be identified by study number in any database. The name and any other identifying details will NOT be included in any electronic file of study data.

Where participants consent, coded genetic data and limited relevant details including, age, gender, information about body habitus, biochemistry, etc can also be made available to collaborators and to the National Institute for Health Research Bioresource (http://bioresource.nihr.ac.uk/), a panel of thousands of volunteers, who are willing to be approached to participate in research studies investigating the links between genes, the environment, health and disease.

## QUALITY ASSURANCE PROCEDURES

The study may be monitored, or audited in accordance with the current approved protocol, Good Clinical Practice, relevant regulations and standard operating procedures.

## ETHICS AND DISSEMINATION
### Declaration of Helsinki

This study will be conducted in accordance with the principles of the Declaration of Helsinki.

### Guidelines for Good Clinical Practice

The Investigator will ensure that this study is conducted in accordance with relevant regulations and with Good Clinical Practice.

### Reporting

The CI shall submit once a year throughout the study, or on request, an Annual Progress report to the REC Committee, HRA (where required) host organisation and Sponsor. In addition, an End of Study notification and final report will be submitted to the same parties.

### Publications

The findings from the study will be published in high-ranking journals in the field and presented at national and international conferences.

## Participant confidentiality

The study staff will ensure that the participants' anonymity is maintained. The participants will be identified only by a participant ID number on all study documents and any electronic database, with the exception of the CRF, where participant initials may be added. All documents will be stored securely and only accessible by study staff and authorised personnel. The study will comply with the Data Protection Act, which requires data to be anonymised as soon as it is practical to do so.

## Expenses and benefits

There will be no payments made to study participants.

## Other ethical considerations

Participants unable to consent for themselves will not be included in the study.

Patients under clinical management for infertility will be approached in a most sensitive manner by our experienced and well-trained team. It is unlikely that our genetic analysis of the participants will reveal anything relevant beyond their normal clinical care so we do not plan to report any such findings to them or their GPs.

## DISCUSSION

This study has been designed to address both uterine fibroids and endometriosis as conditions that both affect the female reproductive system and pose similar problems with regard to pain treatment, fertility and quality of life. By combining the patient collectives into one study, we hope to make use of synergies between the investigations of the two conditions, in addition to the apparent comorbidity between endometriosis patients and those with uterine fibroids.[28 29] Uniquely, the study protocol allows for the sampling of endometrium on a follow-up visit, which will allow for the assessment of the local, molecular effects of treatment within the same participant.

FENOX has been designed with the EPHect principles[14–17] in mind to ensure standardisation and reproducibility, and thus should deliver high-quality datasets that will be useful and comparable between centres. We are currently expanding the collection of samples to sites outside of Oxford, with a view to make FENOX a multicentre study within the UK eventually.

**Contributors** TTT wrote the study protocol with KTZ and CMB. CH, KB and ES consented participants and collected samples. HMN, KSS and TTT processed samples. KG, EJF and CC documented samples and designed electronic questionnaires. SM determined menstrual cycle stages by histology. KTZ and CMB conceived the study.

**Funding** TTT received funding from the Nuffield Benefaction for Medicine and the Wellcome Institutional Strategic Support Fund (ISSF, ref. no. 5258). HMN received funding from the Oxfordshire Health Services Research Committee. KG was supported by a grant from the National Institutes of Health USA (R01HD094842). Funding for this study has been obtained from the Nuffield Department of Women's & Reproductive Health under the Oxford/Bayer-Alliance for Women's Health.

**Competing interests** None declared.

**Patient consent for publication** Not required.

**Ethics approval** Approval for this study has been granted by the South Central—Oxford B Research Ethics Committee (REC No. 17/SC/0664), by the Health Research Authority (HRA) and by the Oxford University Hospitals NHS Foundation Trust. The FENOX protocol was based on the previous ENDOX study (REC reference 09/H0604/58). It was originally approved in January 2018, with amendments approved in April 2019.

**Provenance and peer review** Not commissioned; externally peer reviewed.

**ORCID iD**

Thomas Theodor Tapmeier http://orcid.org/0000-0002-7921-2326

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
