## [Reviewer comments · BMJ Open]

ARTICLE DETAILS

TITLE (PROVISIONAL)	A protocol for a longitudinal, prospective cohort study investigating the biology of uterine fibroids and endometriosis, and patients' quality of life – the FENOX study.
AUTHORS	Tapmeier, Thomas Theodor; Nazri, Hannah Mohamed; Subramaniam, Kavita S; Manek, Sanjiv; Garbutt, Kurtis; Flint, Emma J; Cheuk, Cecilia; Hubbard, Carol; Barrett, Kelly; Shepherd, Emily; Zondervan, Krina T; Becker, Christian Malte

VERSION 1 – REVIEW

REVIEWER	Ellen Løkkegaard Dept. Obstetrics and Gynecology North Zealand Hospital Institute of Clinical Medicine University of Copenhagen Denmark
REVIEW RETURNED	15-Jul-2019

GENERAL COMMENTS	This is a protocol article on the establishment of a biologic biobank including women undergoing gynecological surgery on the indication endometrioses, fibroids or unrelated condition. The abstract is not making the design clear, so the reading of the main text is a careful scrutinization to get the picture. It would have been helpful if the abstract in line 27 included information on women undergoing operative surgery are included. Also it is not clear how the non-affected control arm is generated. How is this arm collected? It should be described somewhere how they are included. Are they just the women found not to have visual disease during surgery? What are requirements to surgery to rule out disease? In line 253 p 8 it is stated women undergoing TCRF are included. It is not very clear from the inclusion criteria's that women undergoing TCRF are also included. Also hysteroscopies are not included in the figure 1. How are myometrical samples taken from women not having disease in the uterus? This is not described. Page 9 line 276 it is stated that women will be asked to contact the study team when the next menstrual period after the procedure started to calculate the length of that cycle. The could be further described in detail so the meaning is clear. In p 10 line 308 it is stated 50 ml blood will be collected however the volumes only add up to 40 ml if the heparin treated samples is assumed to be 2... This is not precisely specified what glasses with what amount. In p 10 line 321 it is stated peritoneal fluid is collected, how?
--

	What is ENDOX short for? P 15 line 468. There is no attempt to do any power calculation. The latest reference is from 2017.
--	--

REVIEWER	Holly R. Harris Fred Hutchinson Cancer Research Center
REVIEW RETURNED	22-Oct-2019

GENERAL COMMENTS	Outcomes Secondary outcomes “The correlation of data and endometriosis status will allow us to define novel biomarkers of the disease.” Can the authors specify what specific types of data (e.g. genetic, biomarkers, etc) they are referring to. Study Design Should it be ‘endometriosis tissue’ and not ‘endometrial tissue’ or ‘endometrial-like tissue’ as an alternative? The ‘control’ group appears to be women ‘undergoing surgery for unrelated gynaecological conditions.’ If these women are found to incidentally have endometriosis or fibroids as a result of the surgery will they then be included in the endometriosis/fibroid group. Further details on the rationale for comparing the endometriosis/fibroids cases to only another surgical group (who themselves may not fully represent “healthy” women) should be included. When the baseline questionnaire is initially introduced it should be clarified if this is completed prior to surgery as it has been done with the blood, urine and saliva collection. What range of time (minutes? hours?) is the ‘lengthy’ baseline and ‘shorter’ version of the questionnaire expected to take. Participants How will participants with both endometriosis and fibroids be handled in recruitment and analysis? For the cancer exclusion does this include non-melanoma skin cancers? Study settings Baseline assessments The researchers describe that “Consented participants will be asked by the research team to complete a baseline questionnaire before their surgery as appropriate for their condition.” Won’t there be some participants who are having a diagnostic surgery? If so, what questionnaire will they complete. Will those with both endometriosis and fibroids complete two sets of questionnaires? Subsequent visits In this section is should be clarified if the additional endometrial sample will be only requested from those with uterine fibroids or to all participants. Sample Handling If sample collection is being collected using WERF EPHect Guidelines these could be cited in the sample handling section. In
---

	particularly, could more details about the peritoneal collection be included or a reference with more details be cited. Interventions Non-clinical Interventions It doesn't seem like usually language (unless determined by the journal requirements?) to consider questionnaires and medical record reviews as interventions. Is there different terminology that could be considered? Reference for EHP-30 should be provided. Description of statistical methods The researchers might consider including what type of multivariable models they may use and what confounders they would consider in the main analyses using these models. The number of participants When the researchers say "Given our current recruitment rate" are they referring to patients who are being seen at the Endometriosis CaRE Centre or are they referring to actual participants consented and enrolled in a prior study. If the latter, they should change the language to from 'recruitment rate' to more accurately describe patients versus study participants. Further, if the latter scenario is true above, then can the authors provide evidence/rationale from their prior Endox study that they will be able to recruit the number of participants described in this section given the study timeline? Analysis of outcome measures Can the 'revised American Fertility Score' have a reference included? What statistical tests or other methods will be used to identify 'participants that have particularly extreme values'? The sentence 'The analysis will be performed on the whole data set' is a bit confusing with where it is located/written given that it directly follows a section that discusses the test set vs training set. Further, the description of what 'influential differences' that will lead to matching should be more clearly described. Do they mean characteristics such as age that are commonly matched on or other characteristics that may not be as common? The way it is currently written is unclear. What statistical test (if any) will be used to determine whether the whole data set is used versus when matching will be utilized. Minor comments Abstract mentions average of 6-9 years before endometriosis diagnosis while the introduction says 8-12 years. The abstract Introduction reads: "with severe menstrual and non-menstrual pain and subfertility the main symptom", should it be symptoms to cover both pain and infertility?
--	---

VERSION 1 – AUTHOR RESPONSE

Reviewer: 1

Reviewer Name: Ellen Løkkegaard

Institution and Country:

Dept. Obstetrics and Gynecology, North Zealand Hospital

Institute of Clinical Medicine, University of Copenhagen, Denmark

Please state any competing interests or state 'None declared': None Declared

Please leave your comments for the authors below

This is a protocol article on the establishment of a biologic biobank including women undergoing gynecological surgery on the indication endometrioses, fibroids or unrelated condition.

The abstract is not making the design clear, so the reading of the main text is a careful scrutinization to get the picture. It would have been helpful if the abstract in line 27 included information on women undergoing operative surgery are included. Also it is not clear how the non-affected control arm is generated. How is this arm collected? It should be described somewhere how they are included. Are they just the women found not to have visual disease during surgery? What are requirements to surgery to rule out disease?

We thank the reviewer for her assessment. We included the “undergoing surgery” in line 29 of the abstract. We added a paragraph on the control group as a limitation into the article summary (line 59): “The control group comprises of women undergoing surgery for gynaecological indications other than endometriosis or uterine fibroids; thus, they are not completely healthy controls. This is a limitation of this study. However, we cannot ethically source tissue samples from healthy individuals.”

In line 253 p 8 it is stated women undergoing TCRF are included. It is not very clear from the inclusion criteria's that women undergoing TCRF are also included. Also hysteroscopies are not included in the figure 1.

We included a new figure 1 to clarify the different procedures included in the study, and hope that this makes it easier to follow the setup of the study groups. The flow chart figure is now figure 2.

How are myometrical samples taken from women not having disease in the uterus? This is not described.

If the uterus is not accessed, we will not take myometrial samples. We will collect only those samples that are accessible during the planned clinical procedure.

Page 9 line 276 it is stated that women will be asked to contact the study team when the next menstrual period after the procedure started to calculate the length of that cycle. This could be further described in detail so the meaning is clear.

We added the following to clarify this (line 296): “Unless participants underwent a hysterectomy, all included women will be asked to contact the study team when the next menstrual period after the procedure started. Together with the last menstrual period (LMP) date given at the time of the procedure, this date will be used to calculate the length of the cycle.”

In p 10 line 308 it is stated 50 ml blood will be collected however the volumes only add up to 40 ml if the heparin treated samples is assumed to be 2... This is not precisely specified what glasses with what amount.

Thank you for pointing this out. We added the missing “2x” to the sentence (line 333): “50 mL. These are divided into (at least) EDTA- (2 x 9 mL) and heparin-treated samples (2 x 6 mL, both anti-coagulation), serum (2 x SST, 5 mL) and and two plain blood samples of 5 mL. The different vials are

colour-coded and frozen at -80°C.”

In p 10 line 321 it is stated peritoneal fluid is collected, how?

We included a sentence from our SOP (line 346): “During surgery, the peritoneal fluid will be collected by the surgeon using a syringe or through mechanical suction on ice. Depending on the volume (up to 15 mL), an aliquot will be centrifuged at 300 g, and the pellet (cells) stored at -80°C for further analysis. The cell-free supernatant will also be stored at -80°C.”

What is ENDOX short for?

We included the extended title of the ENDOX study in line 411: “A study to identify possible biomarkers in women with Endometriosis at Oxford – ENDOX”. The abbreviation itself is drawn together from endometriosis and Oxford.

P 15 line 468. There is no attempt to do any power calculation.

The exploratory nature of our study makes this difficult with regards to the multivariate logistic regression models but we included a paragraph on power calculations for correlations and ANOVA testing in line 508.

The latest reference is from 2017.

The study protocol was originally written in 2017 and approved between January and March 2018. However, we updated the literature with a more recent review on endometriosis.

Reviewer: 2

Reviewer Name: Holly R. Harris

Institution and Country: Fred Hutchinson Cancer Research Center

Please state any competing interests or state ‘None declared’: None declared

Please leave your comments for the authors below

Outcomes

Secondary outcomes

“The correlation of data and endometriosis status will allow us to define novel biomarkers of the disease.” Can the authors specify what specific types of data (e.g. genetic, biomarkers, etc) they are referring to.

We included “cellular, molecular and genetic data” into the sentence (line 126) to describe the type of data we expect to derive from the samples collected.

Study Design

Should it be ‘endometriosis tissue’ and not ‘endometrial tissue’ or ‘endometrial-like tissue’ as an alternative?

Thank you, we changed the wording accordingly.

The ‘control’ group appears to be women ‘undergoing surgery for unrelated gynaecological conditions.’ If these women are found to incidentally have endometriosis or fibroids as a result of the surgery will they then be included in the endometriosis/fibroid group.

No, because they could not be consented in time, fill in the baseline questionnaire etc. – these patients will not be able to be included in the study, unfortunately.

Further details on the rationale for comparing the endometriosis/fibroids cases to only another surgical group (who themselves may not fully represent “healthy” women) should be included.

We included a paragraph on the control group into the study summary (line 59): “The control group comprises of women undergoing surgery for gynaecological indications other than endometriosis or uterine fibroids; thus, they are not completely healthy controls. This is a limitation of this study.

However, we cannot ethically source tissue samples from healthy individuals.”

Patients with both endometriosis and fibroids will enter into the analysis case groups as per the research question asked. We included this into the new Data Analysis Plan section (line 503): “Patients with both endometriosis and fibroids will enter into the analysis according to the research question asked; we will conduct sensitivity analyses on this comorbid group to examine to what extent they influence the results.”

When the baseline questionnaire is initially introduced it should be clarified if this is completed prior to surgery as it has been done with the blood, urine and saliva collection.

We included this information in line 171: “There will be a lengthy questionnaire at baseline before surgery [...]”.

What range of time (minutes? Hours?) is the ‘lengthy’ baseline and ‘shorter’ version of the questionnaire expected to take.

In our experience, the lengthy version takes approximately 45 minutes to complete, the shorter version 30 minutes. We included these estimates into lines 171-174: “There will be a lengthy questionnaire at baseline before surgery (taking an estimated 45 minutes to complete), and shorter versions (taking up to 30 minutes to complete) post-operatively at 6-8 weeks, 6 months, 12 months and thereafter yearly for a total of five years after surgical intervention.”

Participants

How will participants with both endometriosis and fibroids be handled in recruitment and analysis? Patients with both endometriosis and fibroids will be recruited as patients with either of these conditions. The baseline questionnaire allows for both conditions, and we will include them in the analysis as relevant for the research questions asked.

For the cancer exclusion does this include non-melanoma skin cancers?

Yes, any type of cancer.

Study settings

Baseline assessments

The researchers describe that “Consented participants will be asked by the research team to complete a baseline questionnaire before their surgery as appropriate for their condition.” Won't there be some participants who are having a diagnostic surgery? If so, what questionnaire will they complete. Will those with both endometriosis and fibroids complete two sets of questionnaires? We historically used different questionnaires for endometriosis and fibroids but have merged these now, so all patients complete the same questionnaire, which branches at certain questions to include or exclude relevant sections. Participants undergoing diagnostic surgery for suspected endometriosis will complete the endometriosis questionnaire. The questionnaire allows for both conditions so participants with both endometriosis and fibroids will only have to fill in one questionnaire. We removed the phrase “appropriate to their condition” from lines 170 and 245, and amended lines 420 and 423 accordingly.

Subsequent visits

In this section it should be clarified if the additional endometrial sample will be only requested from those with uterine fibroids or to all participants.

This only pertains to participants with uterine fibroids. We amended line 302: “One subsequent visit will be made by participants treated for fibroids who consent to this.”

Sample Handling

If sample collection is being collected using WERF EPHeCT Guidelines these could be cited in the sample handling section. In particular, could more details about the peritoneal collection be included

or a reference with more details be cited.

We cited the WERF EPHEct guidelines (line 316) and included a short description of the collection of peritoneal fluid from our SOP (line 346): “During surgery, the peritoneal fluid will be collected by the surgeon using a syringe or through mechanical suction on ice. Depending on the volume (up to 15 mL), an aliquot will be centrifuged at 300 g, and the pellet (cells) stored at -80°C for further analysis. The cell-free supernatant will also be stored at -80°C.”

Interventions

Non-clinical Interventions

It doesn't seem like usually language (unless determined by the journal requirements?) to consider questionnaires and medical record reviews as interventions. Is there different terminology that could be considered?

The terminology was determined by the reviewing bodies -the research ethics committee (REC) and health research authority (HRA)- when the study protocol was written and approved, and we would thus hesitate to replace these terms.

Reference for EHP-30 should be provided.

We included the reference.

Description of statistical methods

The researchers might consider including what type of multivariable models they may use and what confounders they would consider in the main analyses using these models.

We replaced the short “Description of Statistical Methods” with a more elaborate “Data Analysis Plan” (line 496). However, the details of the analysis will depend on the research question asked. Generally, we will use multivariate logistic regression models in comparisons of endometriosis cases with controls, and fibroid cases with controls, adjusting for confounders relevant to the hypothesis being tested. It is impossible to pre-specify confounders without a research question, as the former is dependent on the latter. However, a priori confounders are likely to be age, ethnicity and menstrual cycle phase.

The number of participants

When the researchers say “Given our current recruitment rate” are they referring to patients who are being seen at the Endometriosis CaRE Centre or are they referring to actual participants consented and enrolled in a prior study. If the latter, they should change the language to from ‘recruitment rate’ to more accurately describe patients versus study participants.

Further, if the latter scenario is true above, then can the authors provide evidence/rationale from their prior Endox study that they will be able to recruit the number of participants described in this section given the study timeline?

In our experience from ENDOX, endometriosis patients are highly motivated to join the research effort and almost all of them enrol in our studies. We changed the wording in line 525: “Given our current patient recruitment rate (endometriosis: 100/year, uterine fibroids, 200/year) we estimate an enrolment of approximately 2 × 1200 women over the course of the study (800 endometriosis patients + 400 non-endometriotic controls, 800 fibroid patients + 400 non-fibroid controls).”

Analysis of outcome measures

Can the ‘revised American Fertility Score’ have a reference included?

Yes, we included the reference to the updated scoring system from 1996 (published in 1997), and also corrected the expanded abbreviation to ‘American Fertility Society’.

What statistical tests or other methods will be used to identify ‘participants that have particularly extreme values’?

We define extreme values e.g. as more than 3 SD of the mean and have included that in the

paragraph (line 553): “This may be selected in a stratified method and exclude patients that have particularly extreme values (e.g. > 3 SD from the mean). Patients not included in the test set will make up the training set. Models will then be built on the training set and assessed for predictability on the test set.

The final analysis will be performed on the whole data set. However, if some influential differences e.g. in BMI or comorbidities are seen, then e.g. the women with endometriosis will be matched to corresponding women without endometriosis, or women with fibroids to women without fibroids, and the analysis based on these matched pairs.” However, the exact method of defining outliers depends on the analysis method used. All methods will be published in full, including the definition of outliers.

The sentence ‘The analysis will be performed on the whole data set’ is a bit confusing with where it is located/written given that it directly follows a section that discusses the test set vs training set. Further, the description of what ‘influential differences’ that will lead to matching should be more clearly described. Do they mean characteristics such as age that are commonly matched on or other characteristics that may not be as common? The way it is currently written is unclear. What statistical test (if any) will be used to determine whether the whole data set is used versus when matching will be utilized.

We will develop the multivariate logistic regression models on a small random set of samples (1/3), then test it on the remaining samples before analysing the whole data set. We will potentially match differences such as BMI or comorbidities. We used cluster analysis before (Rahmioglu et al. 2017, cited in the “Data Analysis Plan”) to look at the distribution of data points, or we will adjust for confounding factors if found in both cases and controls.

Minor comments

Abstract mentions average of 6-9 years before endometriosis diagnosis while the introduction says 8-12 years.

We have corrected the abstract towards the more pessimistic time frame.

The abstract Introduction reads: “with severe menstrual and non-menstrual pain and subfertility the main symptom”, should it be symptoms to cover both pain and infertility?

Thank you for pointing this out, we added the -s to indicate the plural correctly.

VERSION 2 – REVIEW

REVIEWER	Ellen Løkkegaard Department of Obstetrics and Gynecology North Zealand Hospital University of Copenhagen Denmark
REVIEW RETURNED	29-Dec-2019

GENERAL COMMENTS	The review comments have been sufficiently addressed in the revised version of the paper
--

REVIEWER	Holly Harris Fred Hutchinson Cancer Research Center, USA
REVIEW RETURNED	19-Dec-2019

GENERAL COMMENTS	The authors have adequately addressed the majority of my previous comments. One issue is still not completely clear. In the revised text under study design the authors state “An incidental diagnosis of endometriosis or uterine fibroids will lead to the patients’ inclusion into the relevant case groups.” This seems like
--

	an appropriate plan. However, in the prior review I asked “If women in [the control group] are found to incidentally have endometriosis or fibroid as a result of the surgery will they then be included in the endometriosis/fibroid group.” The response to this was “No because they could not be consented in time, fill in the baseline questionnaire, etc. – these patients will not be able to be included in the study, unfortunately.” This seems to contradict the changes made in the text. Can the authors clarify?
--	---

VERSION 2 – AUTHOR RESPONSE

Reviewer: 1

Reviewer Name: Ellen Løkkegaard

Institution and Country:

Department of Obstetrics and Gynecology

North Zealand Hospital

University of Copenhagen

Denmark

Please state any competing interests or state ‘None declared’: None declared

Please leave your comments for the authors below

The review comments have been sufficiently addressed in the revised version of the paper.

Reviewer: 2

Reviewer Name: Holly Harris

Institution and Country: Fred Hutchinson Cancer Research Center, USA

Please state any competing interests or state ‘None declared’: None declared.

Please leave your comments for the authors below

The authors have adequately addressed the majority of my previous comments. One issue is still not completely clear. In the revised text under study design the authors state “An incidental diagnosis of endometriosis or uterine fibroids will lead to the patients’ inclusion into the relevant case groups.” This seems like an appropriate plan. However, in the prior review I asked “If women in [the control group] are found to incidentally have endometriosis or fibroid as a result of the surgery will they then be included in the endometriosis/fibroid group.” The response to this was “No because they could not be consented in time, fill in the baseline questionnaire, etc. – these patients will not be able to be included in the study, unfortunately.” This seems to contradict the changes made in the text. Can the authors clarify?

We thank the reviewer for pointing this out: If patients have been consented and thus the incidental diagnosis is made within the FENOX control group, they will be allocated to the respective case groups. A completely incidentally diagnosis on the other hand, i.e. outside of FENOX, would not allow for consenting and thus inclusion into the study.

VERSION 3 – REVIEW

REVIEWER	Holly Harris Fred Hutchinson Cancer Research Center, USA
REVIEW RETURNED	28-Jan-2020
GENERAL COMMENTS	The authors have addressed all prior comments.